# Prevalence, clinical correlates and outcomes of cardiorenal anemia syndrome among patients with heart failure attending tertiary referral hospital in Dodoma, Tanzania: A protocol of a prospective observational study

**Gidion Edwin**[1,2], **Baraka Alphonce**[2], **Alfred Meremo**[1,2,3], **John Robson Meda**[1,2,4]*

1 Department of Internal Medicine, The University Dodoma, Dodoma, Tanzania, 2 Department of Internal Medicine, The Benjamin Mkapa Hospital, Dodoma, Tanzania, 3 Department of Nephrology, The Benjamin Mkapa Hospital, Dodoma, Tanzania, 4 Department of Cardiology, The Benjamin Mkapa Hospital, Dodoma, Tanzania

* jmedaus@yahoo.com

## Abstract

### Background

Cardiorenal anemia syndrome (CRAS) is a common complication among patients with heart failure and is associated with poor clinical outcomes. However, there is a paucity of published data concerning CRAS, despite of significant increase in heart failure patients attending medical services in developing countries. This study aims to assess the prevalence, clinical correlates, and outcomes of CRAS among patients with heart failure attending the Benjamin Mkapa Hospital in Dodoma, Tanzania.

### Methodology

A prospective observational study is ongoing at the Benjamin Mkapa Hospital in Dodoma, Tanzania. Currently, 92 patients have been recruited into this study and process is not yet completed. The socio-demographic data, clinical correlates, and prevalence of CRAS will be determined at baseline meanwhile, the outcomes of CRAS will be determined during a follow-up period of six months from the date of enrollment. CRAS is the primary outcome of the study. Data will be categorized into CRAS and non-CRAS during statistical analysis. Mean and standard deviation will be used for normally distributed continuous variables while median and interquartile range will be used for skewed data. Frequencies and percentages will summarize categorical variables. Clinical correlates and outcomes of CRAS will be analyzed and compared by using univariate and multivariate logistic regression and Cox proportional hazards models. A two-tailed p-value of less than 0.05 will indicate statistical significance.

**Data Availability Statement:** All relevant data are within the manuscript and its Supporting information files.

**Funding:** The author(s) received no specific funding for this work.

**Competing interests:** The authors have declared that no competing interests exist.

# Introduction

Cardiorenal anemia syndrome (CRAS) is public health concern worldwide among heart failure patients and is associated with poor clinical outcomes [1, 2]. The pathophysiology of cardiorenal anemia syndrome is complex and not clear [2–4]. The reported prevalence of CRAS among heart failure with reduced ejection fraction ranges 4.6% and 35.4% in developed countries [5–7] and 19% to 66% in developing countries [2, 5].

CRAS management poses great challenging due to increased cost of services and disease-drug interactions, which portends poor prognosis [1, 2, 8] which explains a very high CRAS related mortality in developing countries (73.5%) than in developed countries (9.5%) [1, 2, 9]. There are several clinical correlates which have been reported to contribute the development and worsening of CRAS including among others diabetes, hypertension, dyslipidemia, obesity, alcoholism and iron deficiency with or without anemia but all these correlates have not yet been established in our settings [5, 10–12]. The most common outcomes of CRAS include all-cause mortality, poor quality of life, heart failure hospitalization, worsening anemia and worsening renal function [1, 2, 13].

There is a significantly increasing burden of HF patients in Tanzania. In our settings, approximate 20 to 50 patients with heart failure attend daily in respective clinics; however, there is unpublished data from hospital registries pertaining to cardiorenal anemia syndrome [5, 8, 14]. This study is the first to be conducted in our settings and will determine the prevalence, clinical correlates and outcomes of cardiorenal anemia syndrome among the patients with heart failure irrespective of ejection fraction attending the Benjamin Mkapa hospital in Dodoma, Tanzania.

# Materials and methods

## Study objectives

1. To determine the prevalence of cardiorenal anemia syndrome among patients with heart failure attending a tertiary referral hospital in Dodoma, Tanzania

2. To determine the clinical correlates of cardiorenal anemia syndrome among patients with heart failure attending a tertiary hospital in Dodoma, Tanzania

3. To determine the outcomes of cardiorenal anemia syndrome among patients with heart failure attending a tertiary referral hospital in Dodoma, Tanzania

## Study design

A hospital based prospective observational study.

## Study duration

Study is conducted for 9 months period start from 18th August, 2023 to 18th April, 2024. Recruitment is from 18th August, 2023 to 18th October, 2023 and follow-up period will commence a month later after recruitment up to 6 months. Last patient will followed up to 18th April, 2024.

## Study settings

This study is ongoing at the Benjamin Mkapa Hospital (BMH). This is a government tertiary hospital that serves as a national consultant hospital located in the capital city of Tanzania. The hospital is located within the University of Dodoma (UDOM) premises, 13 kilometres from Dodoma city center. It was established in 2015 and named in remembrance of Former third

President, His Excellence Benjamin William Mkapa. This hospital serves a population of Dodoma of about 3.08 million as per Tanzania demographic health survey (TDHS) report published in 2022 [15] and other inhabitants from neighbouring regions of Iringa, Singida, Manyara, Tabora, Morogoro, and Katavi [16]. The hospital has bed capacity of 400. The BMH serves as a teaching hospital for the University of Dodoma for both undergraduate and graduate programs.

## Study population

Patients (Human subjects) with heart failure who met inclusion criteria will be recruited.

**Inclusion criteria.**

1. All patients aged $\geq$ 18 years old

2. Patients with heart failure irrespective of ejection fraction with echocardiographic report within 6-months prior enrollment.

3. Patients with chronic heart failure who will undergo echocardiographic imaging during index contact

4. Patients admitted with a confirmed index diagnosis of acute heart failure

5. Patients with either baseline evidence of chronic kidney disease from medical records within 6-months prior enrollment or new patient with chronic kidney disease based on serum creatinine level during index contact

6. Patient who has signed the informed consent form or proxy consent incase the patient is incapable

**Exclusion criteria.**

1. Patients with known solid or hematological malignancy on chemotherapy or radiotherapy

2. Pregnant women

3. Patients with obvious gastrointestinal bleeding from any cause

4. Patients with established Hemoglobinopathies or other non-gastrointestinal bleeding disorders

5. Patients with known history of Chronic kidney disease stage 5 on iron supplementation, or on renal replacement therapy

6. Patients underwent implantable cardiac devices or pacemaker implantation

## Sample size estimation

To estimate the sample size, the proportionate formula used in prospective cohort studies will be used as it has been adopted from Baraka et al., 2023 [16].

$$n = \frac{\left( Z_{\alpha}/2 \sqrt{\left(\frac{r+1}{r}\right) p * (1 - p*)} + Z_{\beta} \sqrt{\frac{p_1(1-p_1)}{r} p_2(1-p_2)} \right)^2}{(p_1 - p_2)^2}$$

$$p* = \frac{p_2 + r p_1}{r + 1}$$

Where by

r = ratio between the two groups

p1 = CRAS prevalence obtained from literature

p2 = expected prevalence of CRAS from the study

p1 −p2 = effect size

Zβ = standard normal variate for statistical power

Zα/2 = standard normal variate for significance level

The prevalence of CRAS in previous studies was 44.4% [2]

The prevalence of CRAS in this study is expected to be 40%

Therefore;

r = 1

p1 = 44.4%

p2 = 40%

p1 − p2 = 13%

Zβ = 0.84 for statistical power of 80%

Zα/2 = 1.96 for significance level of 95%

n = 226

Considering the 30% attrition rate adopted from previous study [2]

Therefore, the minimum sample size required to meet the objectives will be 300 patients.

## Clinical outcomes

**Primary outcome.** Cardiorenal anemia syndrome (CRAS) is the primary outcome of this study and will be defined as a triad comprising heart failure regardless of ejection fraction, chronic kidney disease with estimated glomerular filtration rate <60mil/min/1.73m$^2$ and anemia of <12 g/dl for women or <13 g/dl for men [1, 2].

**Secondary outcomes.** Secondary outcomes will include:

**All-cause mortality** is any death that will occur within follow-up irrespective of cause [2, 11, 17–19].

**Heart failure hospitalization** is defined as any admission of which the cause-specific is heart failure [2, 3, 5, 6].

**Worsening renal function**: is defined when there is a change in chronic kidney disease stage to lower stage than baseline or a need of a dialysis or both [20, 21].

**Worsening anemia**: is defined when there is a change of hemoglobin level to lower level from baseline which will then need either blood transfusion or erythropoietin use or both [3, 5, 22, 23].

## Study variables

**Objective 1.** Aim to determine the prevalence of cardiorenal anemia syndrome among patients with heart failure irrespective of ejection fraction at baseline Table 1.

**Table 1. Summary and description of objective number 1 variables of the study.**

| Variable | Method of measurement | Operational definition | Level of measurement |
|---|---|---|---|
| CRAS | Patient assessment/medical or echocardiographic records/blood sample | Heart failure with ejection fraction of <50% or >50%, chronic kidney disease with estimated glomerular filtration rate <60mil/min/1.73m$^2$ and anemia with hemoglobin of <13 g/dl for men or <12 g/dl for women | Dichotomous |

**Table 2. Summary and description of variables for the objective number 2.**

| Variable | Method of measurement | Operational definition | Level of measurement |
|---|---|---|---|
| **Age** | Medical records/patient | Age in years | Continuous |
| **Gender** | Medical record/patient report | Male versus female | Dichotomous, categorical |
| **Alcoholic use** | Patient report, interview | Use of alcoholics regardless of amount/units, years or number of drinks per day | Dichotomous, categorical |
| **Anemia** | Blood sample | When hemoglobin level is <13 g/dl for men or <12 g/dl for women | Dichotomous, continous |
| **Iron deficiency** | Blood sample | Serum ferritin of <100μg/l and high sensitive C-reactive protein of ≥3 mg/dl; or serum ferritin of 100 to 300 μg/l and calculated transferittin saturation (TSAT) of < 20% will indicate iron deficiency | Dichotomous, continous |
| **Hypertension** | Medical records/patient report/ patient assessment | Known hypertensive patient from records; or measurement of BP readings by conventional method using digitalized calibrated Blood pressure machine adhered on standard operating procedures and when the average of last two readings taken in between 3 minutes apart is ≥ 140/90mmHg will indicate hypertension | Dichotomous, categorical |
| **Chronic kidney disease (CKD)** | Medical records/patient report | Known patient with CKD from medical records within 6 months retrospectively prior enrollment or when estimated GFR is <60mil/min/1.73m2 by CKD-EPI formula using serum creatinine level | Dichotomous, continous |
| **Dyslipidemia** | Medical records/patient report/blood sample | Evidence of dyslipidemia on medications from records or for unknown lipid status, when the total cholesterol ≥200 mg/dl, low-density lipoprotein cholesterol ≥130 mg/dl, and high-density lipoprotein cholesterol < 40 mg/ dl for men or < 50 mg/ dl for women, and/or serum triglyceride level ≥150 mg/ dl | Dichotomous, categorical |
| **Diabetes** | Medical record/patient report/ blood sample | Known patient with diabetes mellitus on medications from records or for unknown status, is when the random blood glucose is ≥11.1 mmol/L and glycosylated haemoglobin of ≥6.5% | Dichotomous, categorical |
| **Electrolyte imbalance** | Medical report/blood sample | Serum Na+ of <135mmol/L or >145mmol/L and/or serum K+ of < 3.5mmol/L or >5.5mmol/L | Continous |
| **Obesity** | Medical record/patient assessment | When the waist circumference of >94 cm for males or >88 cm for females; or calculated waist-hip ratio (WHR) of >0.9 for men or of >0.8 for women | Dichotomous, categorical |

**Objective 2.** Aims to determine the clinical correlates for the cardiorenal anemia syndrome at baseline among the patients with heart failure irrespective of ejection fraction [2, 6, 24, 25]. Information from medical records such as chronic kidney disease using serum creatinine level and echocardiographic report will be accessed from 18th February, 2023 onwards. Table 2.

**Objective 3.** Aims to determine the outcomes of the cardiorenal anemia syndrome among the patients with heart failure within six months follow up period [2, 5, 7, 11, 25] Table 3.

**Table 3. Summary and description of variables for the objective number 3.**

| Variable | Method of measurement | Operational definition | Level of measurement |
|---|---|---|---|
| **All-cause mortality** | Medical records or close relative through phone contact | Any death occurs within six month follow up irrespective of the cause of death | Dichotomous |
| **Heart failure hospitalization** | Medical record/patient report/ assessment | admission of which cause-specific is heart failure | Continuous |
| **Worsening renal function** | Medical record/patient report/blood sample | When there is a change in Chronic kidney disease stage or a need of dialysis or both | Continuous |
| **Worsening anemia** | Medical record/patient assessment/ blood sample | Change of hemoglobin level to lower level from baseline or a need of blood transfusion or erythropoietin use or both | Continuous |

## Participants' characteristics

Participants are adults of 18 years or older attending internal medicine, cardiology and nephrology departments of the Benjamin Mkapa hospital in Dodoma who met the study criteria [4, 26, 27].

## Data collection process

**Evaluation of the participants.**   A minimum of 300 patients who meet the inclusion criteria will be enrolled in this study from 18[th] August, 2023 to 18[th] April, 2024 including 3-months of enrollment and a 6-month of follow up. Follow-up period will be done on monthly basis via phone calls or through routine clinic visits to assess for the secondary outcomes. Data will be collected using a well-structured questionnaire, which will include socio-demographic information, physical examination and anthropometry, assessment of underlying comorbid conditions, laboratory investigations, and non-invasive cardiac imaging. Socio-demographic data will include age, gender, smoking history, history of alcoholic use either current or previous case, and anthropometric measurements including waist circumference, hip circumference and then, waist-hip ratio will be calculated [28]. Focused clinical history will be exhausted to assess the symptoms related to heart failure using modified Framingham criteria. Two major criteria or one major and two minor criteria will be sufficient to make a diagnosis of heart failure [29]. Subjective Functional status by using New York Heart Association functional criteria will be recorded, which comprises four classes as follows [2, 12, 30]: class 1 means no limitation to physical activities, class 2 means slight limitation to strenuous activities, class 3 means marked limitation to simple and slightly activities and class 4 means symptoms at rest.

**Measurements of clinical variables.**   *Waist circumference.* Is the measurement of the abdominal girth in centimeters and will be measured in erect position with abdomen relaxed and arms aside and will be recorded over the unclothed relaxed abdomen at the smallest diameter between the coastal margin and the iliac crest of which >94 cm for men and >88 cm for women will be defined as central obesity. In addition, hip circumference in centimeter will be recorded over minimal clothing at the level of the greater trochanter (usually the widest diameter around the buttocks). Waist-hip ratio (WHR) is dimensionless measurement and will be calculated from waist circumference and hip circumference. A WHR $\geq 0.9$ or $\geq 0.8$ will indicate obesity. The sensitivity and specificity of waist-hip ratio of more than 85% compared to body mass index (BMI) in predicting obesity [28, 31].

*Blood pressure measurement.* A blood pressure (BP) reading will be measured by the conventional method by using a calibrated automated BP machine keeping with the 2018 AHA/ACC hypertension guideline for standard measurement of BP. The patient should be required to empty the urinary bladder prior to measurement, not to smoke, not take alcoholics or caffeine nor strenuous exercises within 30 minutes prior to measurement. Three BP readings will be taken in relaxed position 3 to 5 minutes apart per each reading and an average of the last two BP readings will be used. Blood pressure $\geq 140/90$ mmHg will be defined as hypertension [2, 16, 32].

**Laboratory and imaging investigations.**   *Collection of blood sample.* The laboratory standard operating procedures accredited and supervised by the Benjamin Mkapa hospital will be warranted throughout the study. Each participant will consent for venipuncture which will inflict pain during sample collection, then laboratory technician will follow the aseptic procedures during sample collection. Before sample collection, collecting tubes (purple toped EDTA (K2/ K3) sodium fluoride plain with additive and another red toped plain tubes, with no additive) will be labeled with the patient's particulars. The tourniquet will be applied 30 to 60

centimeters above elbow joint. The intended site of venipuncture will be cleaned with 70% methylated spirit and the area is left to dry for 15 to 20 seconds before venipuncture. Then 10 mls of venous blood will be taken using a 10-cc syringe.

The tourniquet will be released and removed, and the site of venipuncture will be compressed with a cotton swab for 1 to 5 minutes to arrest bleeding. From the samples, the following tests will be analyzed: lipid profile (high density lipoprotein-cholesterol, low density lipoprotein-cholesterol, total cholesterol and serum triglycerides), full blood picture (haemoglobin level, mean corpuscular volume, mean corpuscular hemoglobin), Iron studies (serum iron, serum ferritin, total iron binding capacity and transferrin saturation), high-sensitive C-reactive protein, and serum creatinine and serum electrolytes (serum sodium and serum potassium). The cool box will be used for transportation of samples to the BMH laboratory for processing 15 to 30 minutes after sample collection. The serum sample will be separated from whole blood by centrifugation at 300 rpm for 5 minutes, and two aliquots will be prepared. A sample is stored at 2˚c—8˚c if the analysis is expected after 2 hours from sample collection especially for iron studies which be analyzed after every 100 collected samples using MAGLUMI 800 CLIA. However, blood samples will be stored at room temperature if the analysis is expected to be done within 2 hours from time of sample collection. Analysis of the sample will be done using clinical chemistry automatic analyzer MAGLUMI 800 CLIA made in China, 2019 and Roche Cobas 6000 CLIA made in USA, 2018.

*Lipid profile*. Dyslipidemia is defined as total serum cholesterol ≥200 mg/dL, or LDL-cholesterol ≥130 mg/dL, or serum triglyceride ≥150 mg/dL, or HDL-cholesterol <40 mg/dL for women and <50 mg/dL for men [16, 33].

*Blood sugar measurement*. Diabetes mellitus will be defined as either known patient with type 1 or 2; and for unknown diabetic status, is when random blood glucose by using blood from finger prick is ≥11.1mmol/L and glycated hemoglobin is ≥6.5% [14, 34].

*Iron studies*. Iron deficiency will be defined as serum ferritin of <100μg/l and high sensitive C-reactive protein of ≥ 3 mg/dl [35], or serum ferritin of 100 to 300 μg/l and transferittin saturation (TSAT) ≤ 20% [7, 10–12, 36].

*Full blood picture*. Anemia will be defined when hemoglobin ≤ 12g/dl for women or ≤ 13g/dl for men. Mean Corpuscular Volume (MCV) and Mean Corpuscular Hemoglobin (MCH) will be recorded to characterize the type of anemia [2, 12, 16].

*Serum electrolytes*. Electrolyte imbalance will be considered when the serum sodium is ≤ 135 mmol/l (hyponatremia) or ≥ 145mmol/l (hypernatremia) and / or potassium level of ≤ 3.5mmol (hypokalemia) or ≥ 5.3 mmol/l (hyperkalemia) [20, 33, 37].

*Blood creatinine*. Chronic kidney disease will be defined following estimation of glomerular filtration rate. Glomerular Filtration Rate will be calculated by using CKD-EPI formula by adjustment of African-American correction factor and value of <60mil/min/1.73m2 will be considered as chronic kidney disease [34, 37, 38].

*Two-dimension transthoracic echocardiogram (2D-TTE)*. Transthoracic Echocardiography (model Vivid TM T9 manufactured by GE Healthcare, USA 2018) will be used for recommended patients who are either newly diagnosed patients or for previous known HF patients of which their echocardiogram reports are of more than 6-months prior to index contact. Transthoracic Echocardiogram will be done by experienced echocardiographer and/or confirmed by consultant cardiologists. Depending on condition of the patient so, a patient will be instructed to undress and expose the chest then, the patient will lie on the left lateral decubitus position and then, change to supine position during imaging for different echocardiogram views. The imaging will take 10 to 20 minutes.

During imaging, the confidentiality and privacy will be warranted. Findings of potential diagnostic importance will be extracted from the echocardiogram report which will include

diastolic dysfunction of either grade, elevated left atrial pressure and left ventricular ejection fraction [26].

**Follow-up procedure.**  Every patient will be followed up on monthly basis within 6-months via phone calls or during regular clinic visit. In case a participant is not reachable following 3 phone calls on 3 alternative days on every fourth week of follow-up month or during clinic visit. Failure to be reachable will indicate lost to follow-up.

## Data management

All collected data will be accessible to research team only. All completed questionnaires will be coded before entered into spreadsheets, and patients will be identified by unique patient' registration number of the BMH linked to patient name, age and sex which will be stored in encrypted secured computer.

## Safety consideration

Throughout this study, the safety of the patients will be highly observed. Only minimal injury and pain is expected during venipuncture. During echocardiogram, the patient needs to undress the chest for procedure so, privacy will be ensured. All patients will receive high quality of care as per national and hospital guidelines despite of his or her readiness to participate in this study. No minors will be enrolled.

## Data analysis

All completed questionnaires will be coded before entered into dataset on Microsoft excel. SPSS windows version 25 program (IBM SPSS, Chicago IL) will be used for data management and analysis [16]. Data will be classified as CRAS or non-CRAS groups. Data will be presented as means ± standard deviations for continuous variables with normal distribution, or median and interquartile range for no-parametric (skewed) variables [1, 2, 13]. Categorical variables will be presented as frequencies and percentages. Socio-demographic variables and clinical correlates of cardiorenal anemia syndrome at baseline will be analyzed and compared using Pearson chi-squared test for the categorical variables, and independent student's t-test for the continuous variables [1, 2, 9]. The clinical correlates including age, gender, alcohol use, smoking history, hypertension, obesity, diabetes, and dyslipidemia and iron deficiency will be analysed and compared by using univariate and multivariate logistic regression. Cox proportional hazards model will be used to analyze and compare the outcomes of CRAS. A two-tailed p—value of less than 0.05 will indicate statistical significance [2, 7, 13].

## Ethical issues

The study complies with ethical principles that have their origin in the Declaration of Helsinki. The ethical clearance to conduct the study is obtained from the institutional research review ethical committee (IRREC) under the Vice chancellor's office of the University of Dodoma given a reference number (MA.84/261/64/123). The approval and permission for data collection is obtained from the Benjamin Mkapa hospital.

## Study timeline

Study duration is 9-months commencing from 18[th] August, 2023 and will end on 18[th] April, 2024.

## Discussion

Globally and in Tanzania, there is significantly step-rise burden of heart failure patients attending medical services and most of these patients succumbs CRAS which then, portends poor clinical outcomes [2, 21]. Reported data of CRAS irrespective of ejection fraction is significantly higher in developing countries as compared to developed countries [1, 2, 21]. In our settings, there is paucity of data concerning CRAS despite of having many patients with heart failure attending medical services. The ongoing prospective observational study is assessing the prevalence, clinical correlates and outcomes of CRAS. During discussion, the findings from this study will be by compared with other previous studies so as to draw up feasible conclusion. Establishment of the burden of CRAS, clinical correlates and outcomes in our settings will be a benchmark towards the large studies with high level of evidence and recommendation in order to improve the health care delivery systems so as to reduce the poor clinical outcomes and hence, improves quality of life.

## Supporting information

**S1 Dataset.**
(XLSX)

## Acknowledgments

The authors will appreciate the patients and staffs of the Benjamin Mkapa Hospital in Dodoma, Tanzania for their readiness and voluntarily active participation in this study.

## Author Contributions

**Conceptualization:** Gidion Edwin, Alfred Meremo, John Robson Meda.

**Data curation:** Gidion Edwin.

**Formal analysis:** Gidion Edwin, Baraka Alphonce.

**Investigation:** Alfred Meremo, John Robson Meda.

**Methodology:** Gidion Edwin, Baraka Alphonce, Alfred Meremo, John Robson Meda.

**Supervision:** Baraka Alphonce, Alfred Meremo, John Robson Meda.

**Writing – original draft:** Gidion Edwin.

**Writing – review & editing:** Baraka Alphonce, Alfred Meremo, John Robson Meda.

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
