## [Editor Report · Decision Letter 0]

8 Dec 2023

PONE-D-23-30946Prevalence, clinical correlates and outcomes of cardiorenal anemia syndrome among patients with heart failure attending tertiary referral hospital in Dodoma, Tanzania:

A protocol of a prospective observational studyPLOS ONE

Dear Dr. Meda,

Thank you for submitting your manuscript to PLOS ONE. After careful consideration, we feel that it has merit but does not fully meet PLOS ONE’s publication criteria as it currently stands. Therefore, we invite you to submit a revised version of the manuscript that addresses the points raised during the review process.

ACADEMIC EDITOR:

Dear Authors,

I have read with great interest your study protocol entitled “Prevalence, clinical correlates and outcomes of cardiorenal anemia syndrome (CRAS) among patients with heart failure attending tertiary referral hospital in Dodoma, Tanzania.”

This subject is relevant, and hopefully, it will contribute to the body of evidence regarding the cardiorenal anemia syndrome in this specific study population, which has not been represented enough in current clinical studies. Likewise, I congratulate you on your efforts to investigate this subject in a prospective manner, which should reduce the bias we encounter with retrospective studies.

As stated by Authors, this study has three objectives including determining the prevalence of CRAS in Dodoma, Tanzania, clinical correlates of CRAS, and outcomes of CRAS. However, the sample size calculation was based on the primary outcome which is CRAS given expected estimates in the formula. As such, it is very likely that you won’t be able to meet enough power for your secondary outcomes, and as such, reliably investigate your third objective which is to determine the outcomes of CRAS. To meet third objective, I would recommend to base the power calculation on both outcomes (CRAS and clinical endpoints for CRAS). To increase the power, you can use a composite endpoint for HF trials (Eur J Heart Fail. 2016 May;18(5):482-9).

We look forward to receiving your revised manuscript.

Kind regards,

Milos Brankovic, MD, PhD, MSc

Academic Editor

PLOS ONE

2. Please include a separate caption for each figure in your manuscript.

---

## [Author Response · Author response to Decision Letter 0]

14 Dec 2023

Response to reviewers

Dear Editor-in-chief,

PLOS ONE.

We express our cordial gratitude for the valuable and constructive assessment and review of our paper. The reviewer’s comments have helped us in deed to further strengthen the quality of our work. With utmost care, we are incorporating all the suggestions/corrections as proposed by the reviewers. 

In the following, we address the concerns point by point:-

Comment 1:-

The PLOS ONE publication guidelines are not followed and adhered.

Response 1: The manuscript has been updated so as, to strictly adhere to the recommended guidelines, including:-

Superscripting of the authors’ affiliations (refer, page number 1, line 9 to 12)

Omission of the full physical address and use of corresponding author’s initials (refer, page number 1, line 15)

Styling, formatting and font size of main document as per PLOS ONE publication guideline (refer, page numbers 5, 12, 16, 18 and 19, lines 104, 106, 195,310, 355, 359)

Omission of incomplete reference list and updating the complete reference list as per guideline (refer, page numbers 20 to 25, lines 377 to 544)

Updating tables’ captions and citation of the tables was done (refer, page numbers 8 to 11, lines 177, 178, 184, 185, 191, 192)

Spacing of words (refer, page number 14, line 259)

Page breaking (refer, page numbers 16, 18 and 19, lines 310, 353, 359)

 

Comment 2:-

Difficulties in assessment of the third objective of the study since the power of the study is not strong enough. Sample size estimation favors the first objective and it will be less likely to assess the third objective.

Response 2: The manuscript was updated as per PLOS ONE publication guideline as follows:-

Re-calculation and re-estimation of sample size and increase on sample size was done so as to strengthen the power of our study. Also, reference article used for estimation of sample size was changed (refer, page numbers 6 and 12, lines 131, 143, 147, 148, 153, 201)

Modification and re-writing of the third objective of the study was done by including all-cause mortality and heart failure hospitalization after passing through composite endpoints for heart failure (HF) trials as per articles recommended by reviewers. Additional referencing was done (refer, page number 7, lines 164 to 172)

Sincerely,

John Meda

Corresponding author.

---

## [Editor Report · Decision Letter 1]

18 Dec 2023

Prevalence, clinical correlates and outcomes of cardiorenal anemia syndrome among patients with heart failure attending tertiary referral hospital in Dodoma, Tanzania:

A protocol of a prospective observational study

PONE-D-23-30946R1

Dear Dr. Meda,

We’re pleased to inform you that your manuscript has been judged scientifically suitable for publication and will be formally accepted for publication once it meets all outstanding technical requirements.

Kind regards,

Milos Brankovic, MD, PhD, MSc

Academic Editor

PLOS ONE

Additional Editor Comments (optional):

Authors have addressed previous comments appropriately. No further comments. 
---

## [Editor Report · Acceptance letter]

22 Dec 2023

PONE-D-23-30946R1 

PLOS ONE

Dear Dr. Meda, 

I'm pleased to inform you that your manuscript has been deemed suitable for publication in PLOS ONE. Congratulations! Your manuscript is now being handed over to our production team.

Kind regards, 

on behalf of

Dr. Milos Brankovic 

Academic Editor

PLOS ONE